# A Simulation Study of the Effect of HCNG Fuel and Injector Hole Number along with a Variation of Fuel Injection Pressure in a Gasoline Engine Converted from Port Injection to Direct Injection

Javad Zareei [1,*], José Ricardo Nuñez Alvarez [2], Yolanda Llosas Albuerne [3], María Rodríguez Gámez [3] and Ángel Rafael Arteaga Linzan [3]

1   Department of Biosystems Engineering, Faculty of Agriculture, Ferdowsi University of Mashhad, Mashhad 9177948974, Iran
2   Energy Department, Universidad de la Costa, Barranquilla 080002, Colombia
3   Department of Mechanical Engineering, Universidad Técnica de Manabí, Portoviejo 130105, Ecuador
*   Correspondence: javadzareei@um.ac.ir

**Abstract:** The number of injector holes and the fuel-injection pressure in an internal combustion engine can affect engine performance and exhaust emissions. Conversion of a port-injection gasoline engine to an HCNG direct-injection engine improves engine performance and exhaust emissions. In addition, increasing the injection pressure helps to increase engine performance. In this study, AVL Fire software was used to perform simulation by certain adjustments. The injection pressure was applied in mods of 15, 20, and 25 bars, the injector holes numbers were 3 and 6, the compression ratio changed from 10:1 to 14:1, and the amount of hydrogen enrichment to natural gas was in mods of 10%, 20%, 30%, and 40%. This paper discusses the items above with regard to power, torque, combustion chamber pressure, fuel conversion efficiency, and exhaust emissions. The result determined that increasing the number of injector holes improves the performance engine and reduces CO emission so that the contour plots confirmed the balanced distribution of temperature and pressure. According to obtained results, maximum engine performance improved from 2.5% to 5% at different speeds and 30% added hydrogen, 25 bar injection pressure, and 6-hole injectors. The amount of CO decreased by approximately 30%, and NOx increased by about 10%.

**Keywords:** injector holes number; compression ratio; injection pressure; hydrogen; natural gas; direct injection engine

## 1. Introduction

An enormous amount of environmental pollution is mainly caused by the automotive sector via use of fossil fuel applications [1]. Vehicle exhaust emissions transmit volatile organic compounds, leads, NOx, and CO, and the origin of the highest emissions is from a low air-to-fuel ratio and low exhaust flow; thus, more attention has been shifted toward the deployment of alternative fuels for environmental protection [2,3]. As a consequence, exhaust emission content concentrations are dependent on engine design, exhaust gas after treatment, operational parameters, injector hole number, fuel additives, and fuel types [4].

One of the essential contributions to the reduction of pollution and petroleum reserve consumption is the use of alternative fuels, such as ethanol, CNG, electricity, hydrogen, biodiesel, etc. [5,6]. A potential substitute for fossil fuel is CNG, due to its lower emission, clean burning nature, economy, and its availability in most countries [7]. The octane number of CNG is approximately 120; thus it has higher knock resistance than gasoline, which is leveraged in engines with higher compression ratios, combustion, and efficiency [8,9]. Due to the low cetane number and high vapour pressure, it is sufficient to obtain HCCI

followed by premixed gaseous combustion; hence, CNG has been particularly utilized in SI engines relative to CI engines [10]. The major drawback of CNG is its lower flame speed in the higher temperature of engine components, despite the fact it can be enhanced by the use of dual fuel engines [11].

The combustion process is highly dependent on fuel-injection parameters, type of injector, and the understanding that fuel spray development is essential for the proper control of the process [12]. Changzhao Jiang and his colleagues showed that the mid-sized 3-hole injector produces a spray that more readily collapses as compared to that of smaller or larger hole diameters [13].

A past study showed that to stabilize HCCI engine operation, hydrogen as a reformed gas has lowered the intake temperature and also improved the start combustion in the cylinder [14,15]. It has been highlighted by a number of studies that the utilization of hydrogen as a mixture with CNG is beneficial due to its fast-burning velocity, small quenching distance, and a low ignition energy [16,17], and its usage has resulted in a decreased exhaust emission, improved fuel combustion, and obtained a reduction in NOx emission [18,19].

To obtain a greater engine performance and to improve the volumetric efficiency of the HCNG engine, direct injection is attainable to substitute for port injection [20,21]. Conducted studies reported that higher combustion efficiency and faster combustion duration is obtained from DI's higher rate of heat release [22,23]. A study showed that a lower hydrocarbon emission was obtained in DI compared to port injection, whereas DI controlled the direct injection timing and can increase CNG performance [24,25]. Generally speaking, the port fuel-injection configuration suffers from many limitations, and as a potential solution for the HCNG SI engine, direct injection shows an overall efficiency that could advance the effect of the injector hole number and reduce the exhaust gas dilution rising in the cylinder temperature [26–28].

An increase in hydrogen to CNG in gasoline engines is responsible for decreasing exhaust emissions and improved engine performance [28–30]. Due to the switching of fuels from gasoline to CNG, the density ratio and injection pressure change accordingly. Subsequently this operation suffers from power loss and encounters drivability issues [31,32]; moreover, the procedure affects the CNG proportion prior to entering the combustion chamber. According to a study, the altering initial injection pressure and temperature influences the effects of the engine performance and results in exhaust gas [33,34].

AVL FIRE™ is a powerful tool for the computational fluid dynamics (CFD) simulation package in the internal combustion engine in the era of new drive technologies, and in the development of major components in the electrified powertrain. It has a wide range of uses that includes the prediction of fuel sprays, ignition, combustion, and engine-out emissions [35–37].

In the previous research, only a few studies have been carried out to investigate the effects of injector hole number with respect to changes in fuel-injection pressure in the combustion chamber; hence, there is no literature on the fuel conversion from gasoline to natural gas to introduce the way in which the injector hole number, with respect to the conversion of a port-injection engine to direct injection with HCNG fuel, will affect engine performance. Hence, for this study, the authors aimed to analyze the effects of injection pressure in different modes of 15, 20, and 25 bars and the injector hole number from 3 holes to 6 holes. Moreover, hydrogen enrichment to natural gas was used in the amount of 10%, 20%, 30%, and 40% in order to study the effects on engine power, torque, combustion chamber pressure, and fuel conversion efficiency by utilizing AVL Fire Software CFD simulation. In addition, in this study the extended CFM model was used for simulating combustion, which involves a two-step chemical mechanism; hence, the mean fuel reaction rate will be calculated.

## 2. Methods and Materials

### 2.1. Simulation Conditions and Engine Specifications

Flow and combustion simulations were performed by using AVL Fire software. In order to simulate the engine, all inlet and outlet ducts were considered from the beginning of the inlet duct to the end of the outlet duct and were modeled at 720 degrees. All boundary conditions were considered with the standard test conditions.

The final shape of the simulated model can be seen in Figures 1 and 2. This model includes part of the air intake manifold, smoke outlet manifold, cylindrical chamber, air valves, and smoke.

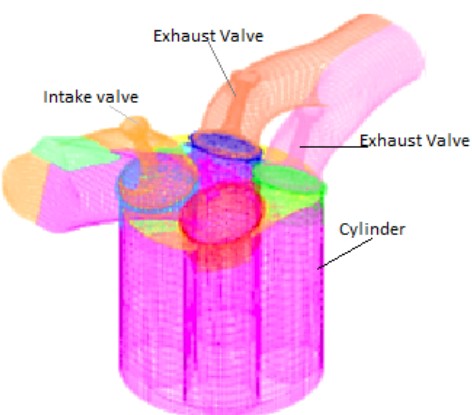

**Figure 1.** Simulated model of combustion chamber with inlet and outlet ducts.

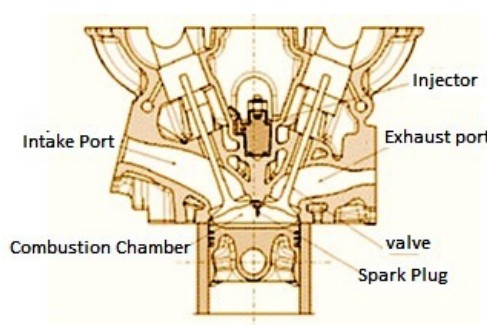

**Figure 2.** The geometrical details of HCNG-DI engine.

The selected engine for this investigation is a four-stroke engine with four cylinders, and a gasoline-type port injection converted to direct-injection mode for the fuel conversion to natural gas and hydrogen fuel. The complete specifications of the engine are given in Table 1.

**Table 1.** Specification of the HCNG-DI engine.

| Engine Parameter | Value | Unit | Engine Parameter | Value | Unit |
|---|---|---|---|---|---|
| Maximum rated power | 82/6000 | kW/rpm | Intake valve opening | 12 | bTDC |
| Maximum rated torque | 148/4000 | Nm/rpm | Intake valve closing | 48 | aBDC |
| Stroke | 84 | mm | Exhaust valve opening | 45 | bBDC |
| Connecting rod length | 131 | mm | Exhaust valve closing | 10 | aTDC |
| Crank radius | 44 | mm | Maximum intake valve lift | 8.1 | Mm |
| Compression ratio | 14:1 | - | Maximum exhaust valve lift | 7.5 | Mm |
| Fuel | | | CNG + Hydrogen | | |

By virtue of the conversion from port injection to direct injection in the internal combustion engine, it is important for the selected injector to be correlatively in accordance with the operating conditions of the engine and impartially in accordance with the changes of the temperature inside the combustion chamber. Hence, the selected injector has been adopted in accordance with the necessary conditions to inject natural gas and hydrogen fuel. The conducive specifications of the injector are in accordance with Figure 3 and Table 2.

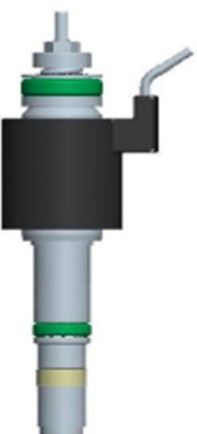

**Figure 3.** View of the injector used to inject gas fuel.

**Table 2.** Specifications of the injector.

| Specification of Injector | |
| --- | --- |
| Opening pressure | 2000 kpa |
| Opening pressure | 1200–2000 kpa |
| Operating voltage | 8–18 V |
| Opening time | 1.5 ms |
| Driver (peak & hold current) | 4 & 2 amp |
| Driver (peak duration) | 2.5 ms |

Fuel Injection

Fuel injection systems have a significant influence on the combustion process and hence a key role to play in improving engine fuel consumption and reducing noxious exhaust emissions. The total injection duration was 50° of the crank angle for achieving a minimum burning time in order to exploit retarded combustion for reduced NOx emissions without loss in efficiency. The compressed (mixture of hydrogen and natural gas) fuel generates a pressure wave (in practical cases), which runs through the pipe and valve, causing the nozzle to open and inject. The pressure wave of the fuel is forced through the system at a speed of approximately 1300 m/s. The task of the injection system is to feed fuel consecutively to each cylinder within a very short period of time. The same amount of fuel is delivered to each stroke. Before converting the engine to direct injection in the case of port fuel injection, gasoline is sprayed into the intake manifold, where it mixes with air, and then is sucked down into the cylinders. Direct injection places an injector on each cylinder, spraying gasoline into the cylinder itself.

*2.2. The Governing Equation*

The survival equation, momentum equation, energy equation, species transfer equation, turbulence equation, combustion model, pollution emission equation, air-to-fuel ratio equation, and particle evaporation equation are some of the relations that are discussed for spark ignition engine simulation. Numerical discretization methods are used to solve the mentioned equations and unions. The accomplished descriptions for fluid flow behavior will be resolved in the solution field through the Cartesian coordinate panel. On the

assumption that the velocity of the fluid flow is in the center of the control volume, it will be defined as a vector. The survival relation is expressed as follows:

$$-\frac{\partial V}{\partial t} = \frac{\partial(\rho u)}{\partial x} + \frac{\partial(\rho v)}{\partial y} + \frac{\partial(\rho w)}{\partial z}. \tag{1}$$

Through the Navier–Stokes equation, the velocity and pressure of a fluid flow change obtained in different dimensions in interdependent,

$$\widehat{\rho}\frac{D\widehat{U_i}}{Dt} = \widehat{\rho}\frac{\partial\widehat{U_i}}{\partial t} + \widehat{\rho U_i}\frac{\partial\widehat{U_i}}{\partial x_j} = \widehat{\rho}g_i + \frac{\partial\widehat{\sigma_{ij}}}{\partial x_j} = \widehat{\rho}g_i - \frac{\partial\widehat{P}}{\partial x_i} + \frac{\partial}{\partial x_j}\left[\mu(\frac{\partial\vartheta_i}{\partial x_j} + \frac{\partial\vartheta_j}{\partial x_i} - \frac{2}{3}\frac{\partial\vartheta_K}{\partial x_K}\delta_{ij})\right], \tag{2}$$

where $\hat{U}i$ is the local velocity of the fluid flow, $\rho$ is the density of the fluid flow, $g_i$ is the acceleration of the earth's gravity, $P$ is the pressure of the fluid flow, $\mu$ is kinematic viscosity, $v_i$ and $v_j$ is the stress tensor between the fluid flow lines, and $\sigma_{ij}$ is the stress resulting from the interaction of the fluid flow with the solution field wall [38].

The enthalpy equation represents the combustion heat outputs, the heat near the walls, and the heat rate released, which affects the performance parameter of the indicator power. The energy equation can be represented as the following continuous sentences:

$$\widehat{\rho}\frac{D\widehat{H}}{Dt} = \widehat{\rho}\left(\frac{\partial\widehat{H}}{\partial t} + \widehat{U_j}\frac{\partial\widehat{H}}{\partial x_j}\right) = \widehat{\rho}\dot{q_g} + \frac{\partial\widehat{P}}{\partial t} + \frac{\partial}{\partial x_i}\left(\widehat{\tau_{ij}}\,\widehat{U_j}\right) + \frac{\partial}{\partial x_i}\left(l\frac{\partial\widehat{T}}{\partial x_j}\right). \tag{3}$$

$H$ is the local enthalpy of the fluid flow, $q_g$ is the heat exchange rate in the gaseous mixture, $T_{ij}$ is the shear stress between the fluid flow lines, 1 is the fuel ratio, and $T$ is the fluid flow temperature [39].

In relation to species transfer, the exact amount of all species, how they mix, penetrate and evaporate the two fluid phases are stated. In describing the reaction kinetics of gas engines, the behavior of the species at each stage of the reaction is estimated based on the species transfer relationship [40]:

$$\frac{\partial}{\partial t}(\rho Y_i) + \nabla(\rho\vec{v}Y_i) = -\nabla\,\vec{J_i} + R_i + S_i, \tag{4}$$

where $Y_i$ represents the introduced species, $v$ represents the viscosity of the fluid flow, $J_i$ determines the infiltration of the species, $R_i$ represents the rate of production of the species after the reaction, and $S_i$ represents the source of the species created in the previous reaction and enters the new equation.

Determining the air-to-fuel ratio in spark ignition engines determines whether the fuel mixture is rich or dilute. The ratio of air to fuel stoichiometry in a gas engine is considered to be about 14:1. In the research, the ratio of air to fuel is stoichiometric 15:1,

$$AFR = \dot{m}_H{\cdot}LHV_H/\dot{m}_{CNG} \cdot LHV_{CNG} + \dot{m}_H \cdot LHV_H, \tag{5}$$

where $m_{CNG}$ fuel and mHydrogen fuel are the mass flow rates of gaseous fuels and spray fuels, respectively. The ratio of gas fuel flow rate to hydrogen fuel determines the volume percentage of fuels.

### 2.3. Ignition Modelling

In this work, our intent is to consider exhaust emissions related to hydrogen enrichment. Consequently, two kinds of ignition must be modelled: the spark ignition (AKTIM model), and the auto-ignition at the origin of the knock phenomenon. These models are embedded in an ECFM3Z model framework, for the modelling of classical combustion processes (diffusion or propagation flame, using a dedicated laminar flame speed correlation) use. The first step of the present computational study was to find a laminar flame speed correlation that will be well suited to represent the combined combustion of methane

and hydrogen for various volume ratios. Most correlations for the combustion of light hydrocarbons are mostly based on the one that has been first developed by Methgalchi and Keck for the mixture of air with propane, methanol, isooctane, or indolence [41].

### 2.4. Fuel Chemical Equation and Engine Specifications

AVL Fire software has the ability to study combustion by using chemical mechanisms, and has been used in 3D simulations. In most studies, gas engine combustion engine simulation has been performed with the help of AVL FIRE-CHMKIN coupling. With the help of AVL FIRE-CHMKIN coupling, the kinetics of the gas engine combustion engine is considered as the initial condition of the species. The kinetics of 42 species and 168 3.0 mech GRI mechanism reactions were used in this study [42–44]. Table 3. Show the initial and boundary conditions of the solution field for the engine studied in this research.

**Table 3.** Initial and boundary conditions of the solution field [45].

| | |
|---|---|
| Start of injection timing | 19 BTDC (704 to 720° of Crank angle) |
| Intake valve closed | 48 ATDC (600° of Crank angle) |
| Exhaust valve opened | 45 BTDC (836° of Crank angle) |
| Injection pressure | 20 bar |
| Temperature at IVC | 360 K |
| Liquid temperature | 353 K |
| Turbulent kinetic energy | 10 |
| Dissipation rate | 1732.05 |
| Cylinder head | Wall temperature 590 K |
| Piston | Mesh movement temperature 600 K |
| Liner | Wall temperature 580 K (Heat flux = 0) |
| Axis | Symmetry |
| Segment cut | KPeriodic inlet/outlet |

To create suitable conditions in the combustion process, the kinetic dissipation (K-e) model of two equations was used, and the combustion model was selected as the Eddy breakup model. The Dukowicz model was used to analyse the evaporation and infiltration conditions of the species and the Knox emission model was estimated based on the Heywood original model. Other contaminants were also considered as chemical reaction species and their mass changes from the beginning to the end of the simulation are estimated as independent species.

The fuel entering the combustion chamber, which is a combination of natural gas with hydrogen, combines with air. The reaction equation formed in the combustion chamber is as follows from Equation (6),

$$\varepsilon \cdot \varphi (v_1 CH_4(g) + v_2 H_2(g))_{(T_f P_f)} + (0.21 O_2 + 0.79 N_2 + \varpi H_2 O)_{(T_i, P_i)}$$
$$\to (0.21 O_2 + 0.79 N_2 + \varpi H_2 O)_{(T,P)} + \varepsilon \cdot \varphi \cdot (v_1 CH_4(g) + v_2 H_2(g))_{(T,P)},$$

(6)

where $\varepsilon$ is the total mole number of fuel, $\varphi$ represents the fuel/air equivalence ratio, $\varpi$ represents the molar humidity ratio, and $v_i$ stands for the mole amount of component *i* per mole of fuel mixture. If pure natural gas is used, $v_2 = 0$, but $v_1 = 0$ if pure hydrogen is used.

The chemical formulas for combustion of stoichiometric hydrogen–air and methane–air mixture are as follows:

$$CH_4 + 2(O_2 + 3.76 N_2) = CO_2 + 2H_2O + 2 \times 3.76 N_2$$

(7)

$$H_2 + 0.5(O_2 + 3.76 N_2) = H_2O + 0.5 \times 3.762 N_2.$$

(8)

The engine operating conditions for the baseline condition of CFD simulation were chosen at a fixed speed of 1000, 2000, 3000, 4000, 5000, and 6000 rpm. The investigated engine operating conditions covered certain variations in the intake temperature, injection timing, injection duration, and spark ignition timing. The extended CFM model used for simulating the combustion, which uses a two-step chemical mechanism; hence, the mean fuel reaction rate will be calculated. The form of these chemical kinetic reactions is shown in Equations (9) and (10) as below:

$$C_nH_mO_k + \left(n + \frac{m}{4} - \frac{k}{2}\right)O_2 \rightarrow n\,CO_2 + \frac{m}{2}H_2O \tag{9}$$

$$C_nH_mO_k + \left(\frac{n}{2} - \frac{k}{2}\right)O_2 \rightarrow n\,CO + \frac{m}{2}H_2 \ . \tag{10}$$

In these equations, *n*, *m*, and *k* are the number of carbon, hydrogen, and oxygen atoms in the fuel mixture, respectively. When simulating a special fuel mixture that does not exist in the default fuel library of the AVL Fire, some of the mixture's special properties, like LHV and the heat of formation and mass fraction of the different components, the details of which are shown in the Table 2, must imperatively be defined for the software.

### 2.5. Formation of CO and NOx

A. Function of CO formation. It is given by the formation rate of CO from the three-step global reaction scheme of methane in the unit of g mol/cm$^3$s, and its equation can be written as (Borman & Ragland 1998):

$$\frac{d[CO]}{dt} = -1 \times 10^{14.6} exp\left[\frac{-40.000}{RT}\right][CO]^{1.0}[O_2]^{0.25}[H_2O] + 5.0 \times 10^8 exp\left[\frac{-40.000}{RT}\right][CO_2]^{1.0} \tag{11}$$

where *t* is temperature in Kelvin and *R* is the universal gas constant.

B. Function of NO formation. This is determined by the formation rate of NO from the three-step extended Zeldovich mechanism in the unit of g mol/cm$^3$s, and its equation can be stated as (Borman & Ragland 1998)

$$\begin{aligned}\frac{d[NO]}{dt} &= 1.8 \times 10^{14} exp\left\lceil\frac{-38370}{T}\right\rceil[N_2][O] - 3.8 \times 10^{13} exp\left\lceil\frac{-425}{T}\right\rceil[NO][N] \\ &+ 1.8 \times 10^{10} exp\left\lceil\frac{-4680}{T}\right\rceil[N][O_2] - 3.8 \times 10^9 exp\left\lceil\frac{-20820}{T}\right\rceil[NO][O] \\ &+ 7.1 \times 10^{13} exp\left\lceil\frac{-450}{T}\right\rceil[N][OH] - 1.7 \times 10^{14} exp\left\lceil\frac{-24560}{T}\right\rceil[NO][H]\end{aligned} \tag{12}$$

### 2.6. Engine Performance Characteristics

The CFD simulation was executed by defining the events for the engine cycle that started from the crank angle degree of 0° CA by defining the value of the initial pressure and temperature. The simulation finished at the crank angle degree of the top dead center, where the exhaust valves will be opening. The measured intake temperature from the experimental work has been implemented to the engine computational mesh, as the piezo static pressure boundary condition. The injection and ignition timings were adjusted appropriately according to increasing engine speeds. The engine operating conditions are shown in Table 4.

Table 4 shows some properties of hydrogen and methane (relative to natural gas properties) fuel under stoichiometric conditions. In addition, Table 5 shows the energy and mass composition of each type of fuel used in the investigation.

**Table 4.** A number of performance characteristics at different engine speeds.

| Engine Parameters and Unit | Value | | |
|:---|:---:|:---:|:---:|
| Engine speed (rpm) | 2000 | 4000 | 6000 |
| CNG mass(mg) | 5.2 | 5.2 | 5.2 |
| Equivalence ratio | 1.0 | 1.0 | 1.0 |
| Intake port temperature (K) | 305 | 305 | 306 |
| Intake port pressure (bar) | 1.04 | 1.02 | 0.9 |
| Start of injection timing (bTDC) | 130 | 170 | 210 |
| End of injection timing (bTDC) | 80 | 120 | 160 |
| Spark ignition timing (bTDC) | 19 | 23 | 28 |
| Injection pressure (bar) | 20 | 20 | 20 |

**Table 5.** Energy and mass composition of $H_2$-NG fuel.

| | CNG | 10% HCNG | 20% HCNG | 30% HCNG | 40% HCNG |
|:---|:---:|:---:|:---:|:---:|:---:|
| $H_2$ (% Mass) | 0 | 1.21 | 2.69 | 4.52 | 6.72 |
| $H_2$ (% energy) | 0 | 3.09 | 6.68 | 10.49 | 15.59 |
| LHV (MJ/Kg) | 46.28 | 47.17 | 48.26 | 49.61 | 51.41 |
| LHV stoich. mixture (MJ/NM$^3$) | 3.376 | 3.359 | 3.353 | 3.349 | 3.344 |
| CNG mass (mg) | 5.2 | 5.13708 | 5.06012 | 4.96496 | 4.855 |
| Hydrogen mass (mg) | 0 | 0.06292 | 0.13988 | 0.23504 | 0.345 |

To calculate the values in Table 5, the combustion equations must be solved so that the volume or weight values of the combination of natural gas and hydrogen fuel can be obtained correctly. Table 6 shows some of the properties of hydrogen fuel and natural gas. The following specifications indicate that the minimum energy required for the combustion of hydrogen fuel is very low compared to natural gas fuel. Therefore, more attention is required in the maintenance of hydrogen fuel as well as the energy contained in hydrogen fuel.

**Table 6.** Properties of hydrogen and methane fuel under stoichiometric condition [44].

| Properties | Hydrogen | Methane | Unit |
|:---|:---:|:---:|:---:|
| Flammability limits | 4–75 | 5–15 | Vol.% |
| Minimum ignition energy | 0.02 | 0.29 | mJ |
| Flame temperature | 2045 | 1875 | °C |
| Auto ignition temperature | 585 | 540 | °C |
| Diffusion coefficient | 0.61 | 0.20 | $10^{-3}$ m$^2$/s |
| Maximum velocity of flame | 3.46 | 0.43 | m/s |
| Density | 0.65 | 0.08 | kg/m$^3$ |

*2.7. Validation and Statistical Method*

2.7.1. Validation of the Model Using Experimental Results

Modeling of internal combustion engines is an attractive and extensive topic. Accurate three-dimensional modeling of the engine is practically very complex, costly, and time consuming; model validation is possible using various methods. To validate the simulation model, by comparing the data between the real system and the simulation model under the same conditions, when the real system or the scenery system is available, it determines the stability of the test component. In this research, simulation and laboratory data have been used to validate the utilised model. Validation is performed in order to adapt the simulation to the principles of numerical analysis and laboratory results. Validation performed in this

study includes validation of network independence and validation of model compliance with laboratory results.

Appropriate cell modeling simulations were performed several times. Initially, the model entered the dissolved cycle with 800,000 cells (average size per cell 22 mm) and combustion pressure values were considered as output. By reducing the average cell size, the amount of space produced up to 1 million cells and reported improved output results. With the increase of cells to 1.1 million cells, the output results of combustion pressure diverged and went out of the optimal state. Therefore, the best network used in this study had 1 million cells. To validate the simulation, a comparison was made between the laboratory results and the computer model. There is always a numerical gap between the modeling results and the laboratory results, which should be reported as an error between the laboratory results and the computer modeling. In the study, due to the use of two different speeds, it is necessary to validate the simulation conditions with laboratory results in two rounds. In most studies, the combustion pressure parameter is used to validate the models. Comparison of simulation and laboratory results showed that the percentage of error between laboratory results and simulation results is about 6%. Therefore, the outputs of the simulation model are in good compliance with the laboratory results [45]. Figure 4 shows the results of laboratory validation and simulation. The results show 100% in the conditions of using natural gas.

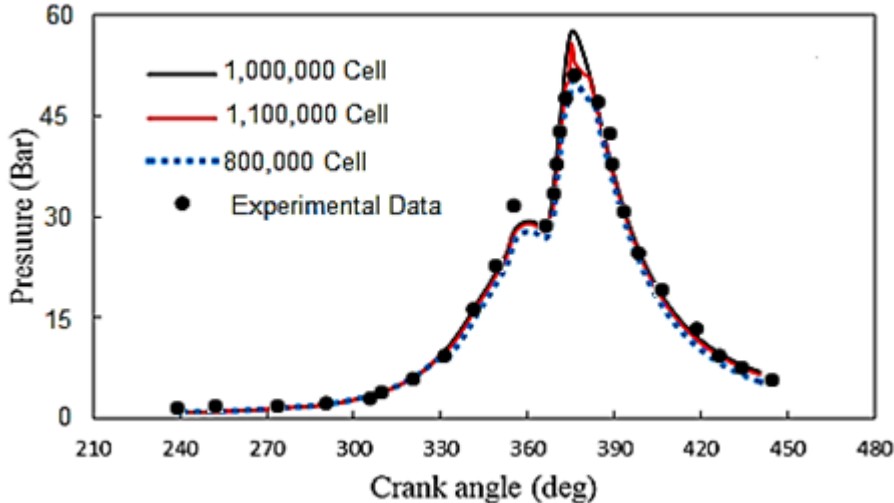

**Figure 4.** Comparison of simulation results with laboratory for model validation.

In addition, in this study the simulation model validated experimental results with 10% HCNG on the engine performance and exhaust emissions. To evaluate the accuracy of the simulation data, a comparison was made between the simulation and experimental data. Figure 5 demonstrates that there is an agreement between simulation results and experimental data. As can be seen, in all cases there is a slight difference between the relationship of these two datasets and the best fitting line ($y = 1.05x + 0.00$, $R^2 = 0.99$). In all cases, the coefficient of determination ($R^2$) between experimental and simulation data is greater than 99%. Therefore, the results from the modeled engine at operating conditions are completely consistent with experimental conditions, indicating the reliability of simulation data.

In the figures below, the curves of (a), (b), (c), and (d) are, respectively, thermal efficiency, power, HC, and CO.

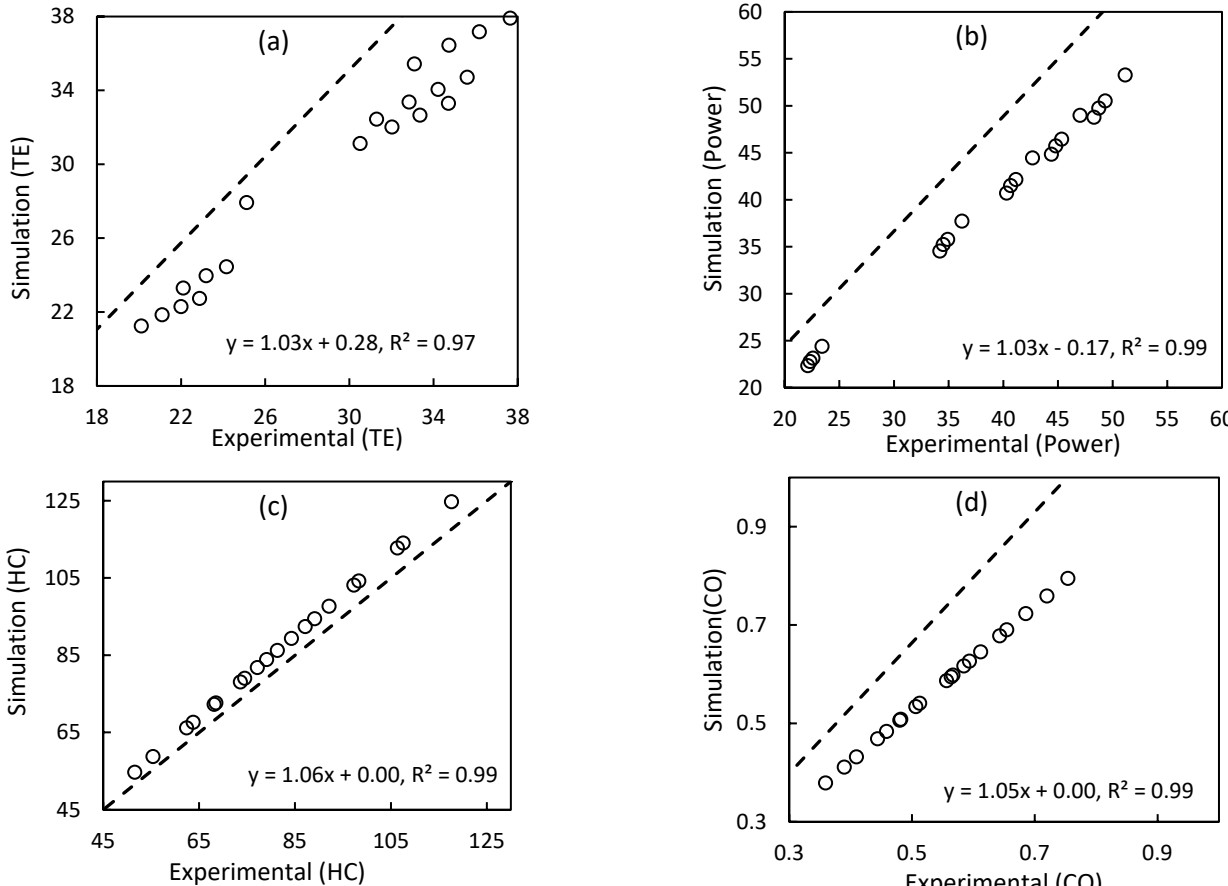

**Figure 5.** The Comparison of simulation results and experimental data at 10% HCNG for model validation.

2.7.2. Statistical Method

For survey and comparison, the effect of fuel blend at speeds of 2000, 3000, 4000, 5000, and 6000 RPM on engine performance and engine exhaust gas emissions of a factorial experimental design were used based on a completely randomized design. In this study, the hydrogen content in five levels includes 0, 10, 20, 30, and 40%, Injection pressure at three levels—15, 20, and 25 bars—and the injector holes number used had values of 3 and 6.

The result of analysis of variance (ANOVA) with aid of *F*-statistic used for evaluation of null hypothesis ($H_0$). For example, $H_0$ indicates that the average engine performance is the same at five levels of hydrogen utilization Equation (4),

$$H_0 : \mu_{0\%} = \mu_{10\%} = \mu_{20\%} = \mu_{30\%} = \mu_{40\%}. \tag{13}$$

If the *p*-value is less than 1% or 5%, then the effect of hydrogen application on changes in engine performance parameters will be significant, and $H_0$ will be rejected. Then pairwise comparisons and performance parameters including braking power, braking thermal efficiency, braking specific fuel consumption, in-cylinder pressure, unburned hydrocarbons, carbon monoxide, and NOx by comparing Tukey's average at 1% level of significance (*p*-value $\leq 0.01$) were compared.

## 3. Results and Discussion

### 3.1. Statical Results

Effect of Injector Holes Number, Injection Pressure on Engine Performance and Exhaust Emissions

In the present section, various engine variables have been analyzed according to statistical models. Taking into account that in the previous section, the validation of the simulation data has been recognized, in this section the engine output results (i.e., performance parameters and engine output pollutants), have been statistically examined.

In Table 7, the results of analysis of variance (ANOVA) of the factorial experimental design is to investigate the significant effects of the three variables of engine speed (rpm), in accordance of the percentage of hydrogen fuel consumption (H), injection pressure (IP), and injector hole number (IHN) by virtue of five (performance power) variables on torque, specific fuel consumption (FCE), Nox, and CO emission.

The values in the total squares (SS) table shows the analysis of variance and the symbols *, **, and ns show the significance of the factors at the level of 1% and 5%, respectively, or non-significance. As can be seen in Table 7, the effects of all four independent variables—rpm, H, IP, and IHN—on the engine performance are at the level of 1%. The increase of the hydrogen percentage in the fuel composition and the injection pressure leads to complete combustion of the fuel due to fast burning and high flame speed. These subjects cause the combustion process to improve and the engine performance to increase. The interaction of two variables of engine speed (rpm) and hydrogen percentage (H) was not dependable for changes in engine power only. Although the interaction of two variables was fascinating at the level of 1%, the interaction of rpm × IHN, H × IP, and H × IHN was significant for two of the engine pollution variables at the level of 1%. The interaction of IP × IHN except FCE on engine performance variables was significant.

**Table 7.** ANOVA analysis of factors affecting engine performance by using factorial experimental design.

| Source | DF | Power | Torque | FCE | NOx | CO (%Vol) |
|---|---|---|---|---|---|---|
| Rpm | 4 | 47086.9 ** | 2517.56 ** | 70.42 ** | 368551 ** | 0.48579 ** |
| H | 4 | 141 ** | 104.02 ** | 16.23 ** | 166658 ** | 1.4675 ** |
| IP | 2 | 237.1 ** | 1084.58 ** | 15.05 ** | 69336 ** | 0.02978 ** |
| IHN | 1 | 11.6 ** | 128.79 ** | 1.73 ** | 23711 ** | 0.01053 ** |
| rpm × H | 16 | 5.6 ns | 10.84 ** | 0.92 * | 1347 ** | 0.01619 ** |
| rpm × IP | 8 | 52.4 ** | 26.83 ** | 1.93 ** | 190 ** | 0.00038 ** |
| rpm × IHN | 4 | 1.40 ns | 2.36 ns | 0.1 ns | 126 ** | 0.00016 ** |
| H × IP | 8 | 0.10 ns | 0.20 ns | 0.00 ns | 111 ** | 0.00101 ** |
| H × IHN | 4 | 0.00 ns | 0.27 ns | 0.00 ns | 38 ** | 0.00036 ** |
| IP × IHN | 2 | 75.5 ** | 196.65 ** | 66.31 ns | 2604 ** | 0.00024 ** |
| Error | 96 | 26.4 | 27.68 | 2.45 | 15 | 0.00003 |
| Total | 149 | 47638 | 4099.8 | 175.1 | $6 \times 10^5$ | 2.01197 |

ns, not significant; *, ** significant at 1% and 5% levels.

Figure 6 shows the compared results of the average engine performance parameters against different engine speeds by the Tukey method at the 5% level. As can be seen, power, FCE, NOx, and CO have a significant upward trend with increasing engine speed. Torque has an upward trend to 4000 rpm and then decreases. The alphabet of "a", "b", "c", "d" and "e" show the difference between results in each figure.

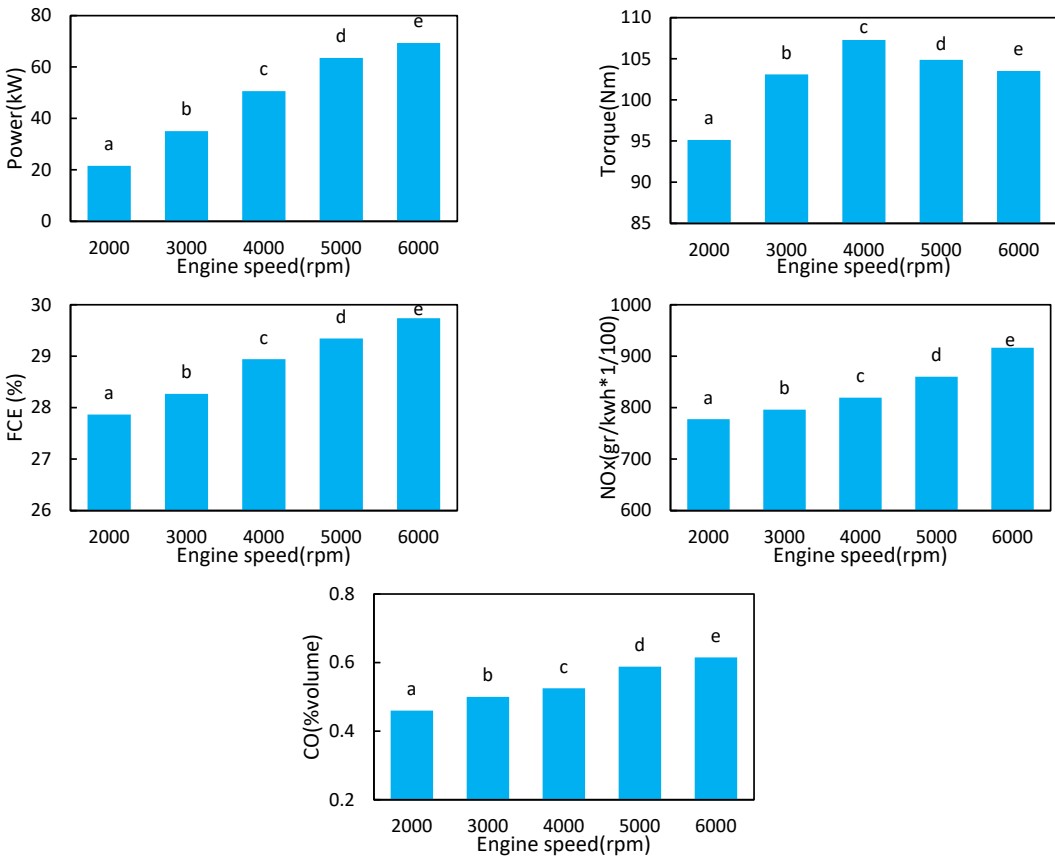

**Figure 6.** The result of comparing the average changes of engine performance parameters against the engine speed factor at a significant level of 5%.

Figure 7 shows the comparison of the average engine performance parameters against the percentage of hydrogen fuel by Tukey method at the level of 5%. As determined, by increasing the percentage of hydrogen up to 20%, the changes in power and torque trend significantly upward, but after that the changes remain nearly constant. Increasing the percentage of hydrogen generates a significant increase in FCE. The early increase of pressure inside the combustion chamber is due to the high speed of hydrogen reaction, which has reduced the amount of power and also reduced the engine torque when the hydrogen amount is more than 30%. This reduces in cylinder pressure on the piston in the expansion cycle and ultimately reduces the engine power. The changes in NOx versus an increase in the percentage of hydrogen fuel trend significantly upward. In contrast, increasing the percentage of hydrogen significantly reduced the CO pollution. The high NOx emissions are due to the high temperature of the combustion chamber gases. Because increasing the percentage of hydrogen increases the temperature of the combustion chamber, it can be concluded that CO decreases. The maximum NOx and minimum CO in 40% hydrogen are obtained in the fuel composition. At the same time, according to the obtained values, improving the operating conditions of the engine as well as the exhaust pollutants proved that the amount of 30% hydrogen creates more suitable conditions in the engine. The alphabet of "a", "b", "c", "d" and e show the difference between results in each figure.

Figure 8 demonstrates the result of the average performance parameters of the engine versus fuel injection pressure (IP) compared by Tukey method at the level of 5%. By the increase of IP, power, torque, and specific fuel consumption the results show a significant upward trend. With the increase of IP, a significant reduction appears and generates advanced changes on the NOx and CO, respectively. By increasing the injection pressure, the fuel combustion efficiency increases and thus the engine performance increases. At

the same time, the combustion temperature also increases. Therefore, NOx increases and the amount of CO decreases. As a general result, it can be considered that increasing the injection pressure improves the engine performance and also reduces the amount of CO output of the engine. The alphabet of "a", "b", and "c" show the difference between results in each figure.

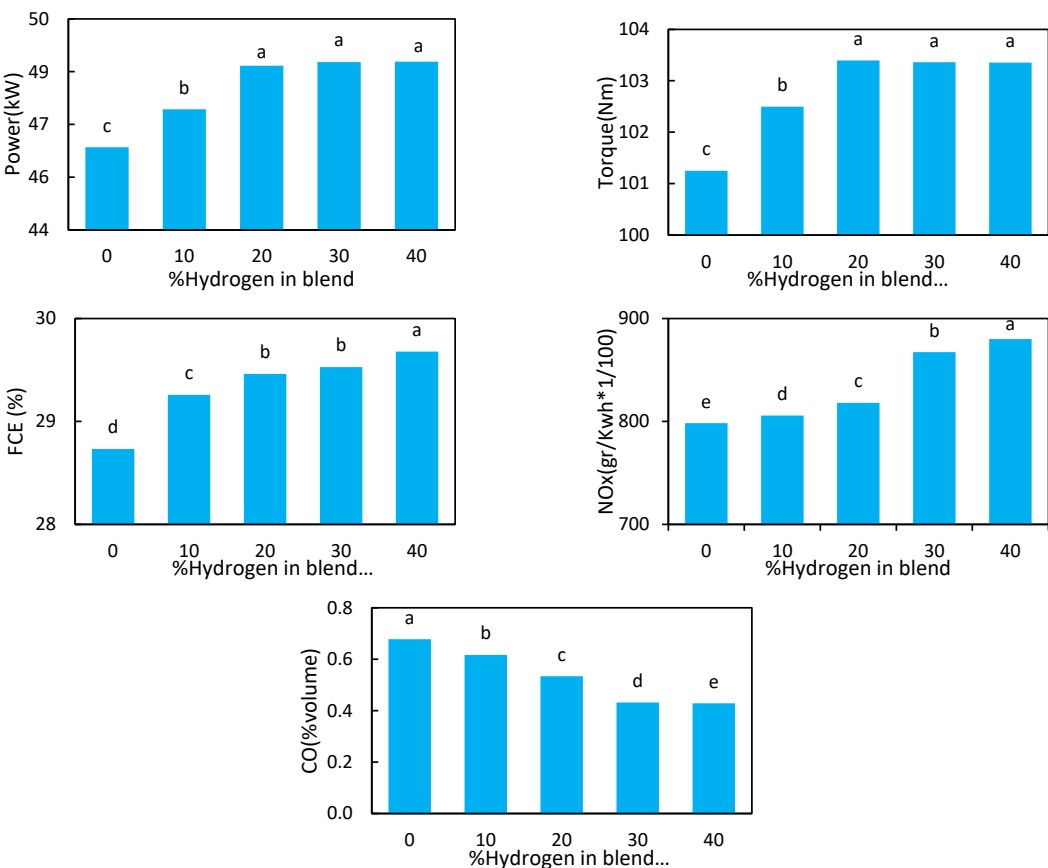

**Figure 7.** The result of comparing the mean changes of engine performance parameters against the hydrogen fuel percentage factor at a significant level 5%.

Figure 9 shows the result of average performance of the engine versus injector hole number (IHN) by the Tukey method at the 5% level. As shown, by increasing IHN, the power, torque, and specific fuel consumption form is significantly decreased. The increase of IHN has significantly reduced and generated advanced changes on the NOx and CO. Decreasing the injector hole number increases the pressure inside the combustion chamber and decreases the combustion delay. This increases the efficiency of the engine and also the performance of the engine such as torque, power, and fuel efficiency and also reduces the amount of CO. The increase in the combustion temperature in the engine cylinder chamber causes a sharp increase in the oxidation rate of CO emissions. Increasing the temperature and increasing the oxidation rate will increase the $CO_2$ and therefore it leads to a decrease in CO. Therefore, the following results show that due to the decrease in the compression ratio and the increase in the amount of NOx in the engine is the only defect, this defect can be eliminated by using various methods such as EGR or catalyst. The alphabet of "a" and "b" show the difference between results in each figure.

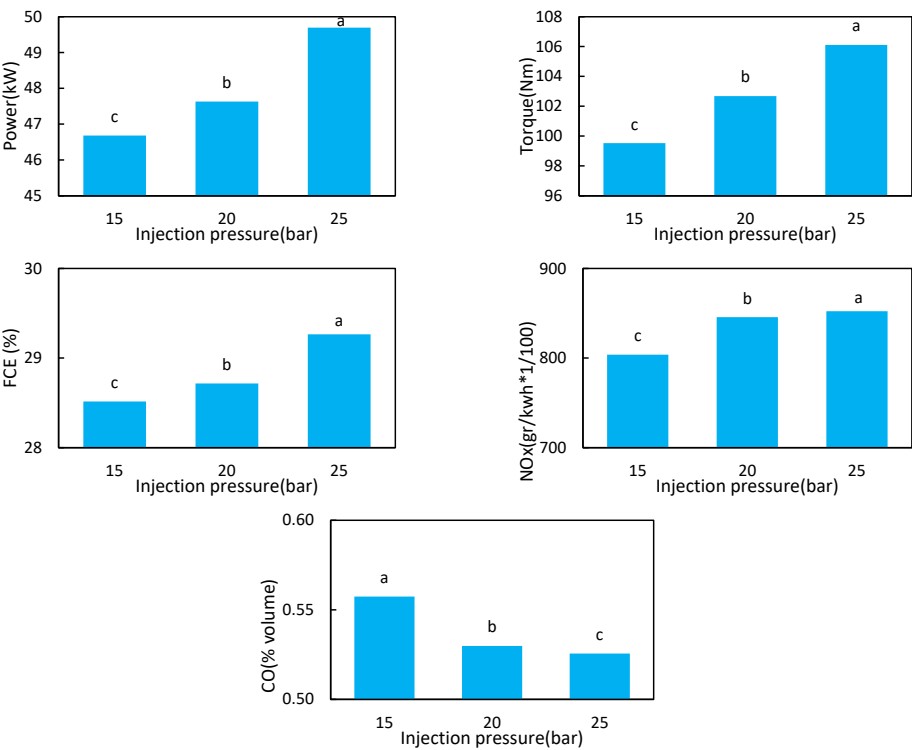

**Figure 8.** The result of comparing the average changes of engine performance parameters against injection pressure (IP) at a significant level of 5%.

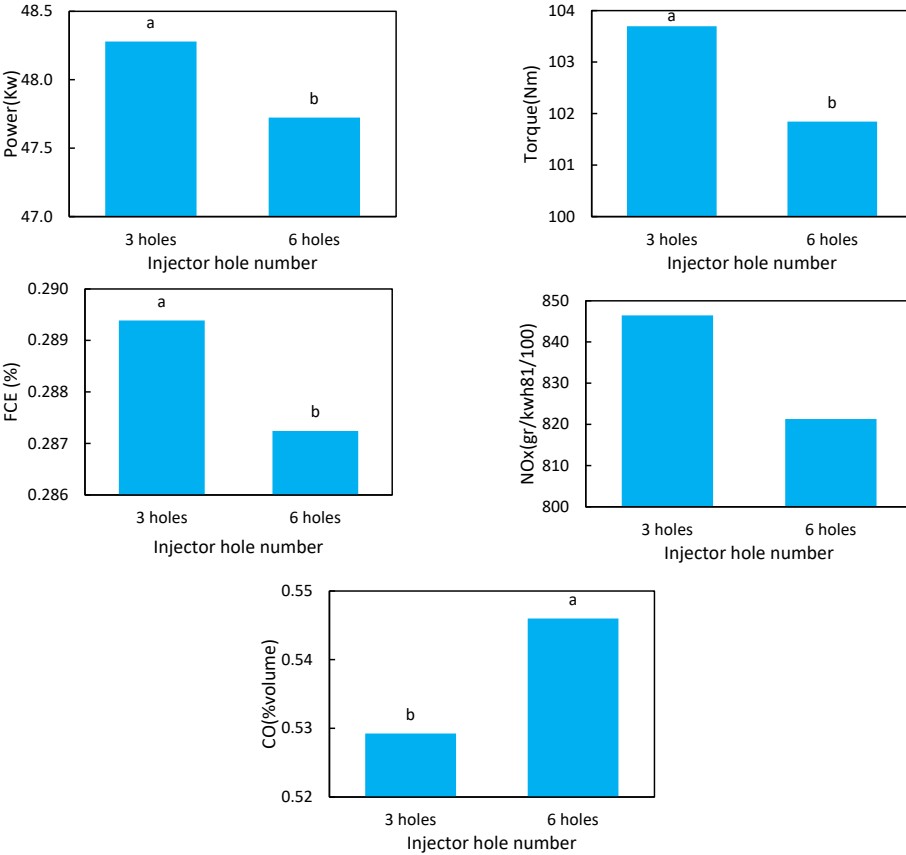

**Figure 9.** The result of comparing the mean changes in engine performance parameters versus injector hole number (IHN) at a significant level of 5%.

### 3.2. Simulation Results

3.2.1. Effect of Injector Holes Number, Injection Pressure on Engine Performance and Exhaust Emissions

The Contour Plot of Pressure and Temperature in Cylinder for 3-Hole and 6-Hole Injectors

The contour plot of in-cylinder pressure for a 3-hole injector and a 6-hole injector during the combustion process in order to characterize the behavior of combustion can be seen in Figure 10 at crank angle positions of 360° and 370° in the expansion stroke at TDC and 10° after it. The maximum pressure for the 3-hole and the 6-hole injector occurs within the engine cylinder is approximately 3.34 MPa and 3.21 MPa, respectively. The maximum pressure value is reached at the top part of cylinder head when the piston touches the TDC position during compression stroke. It also can be noticed that the peak pressure tends to appear with the 3-hole injector. Therefore, it can be said that increasing the number of holes in the injector causes the movement of the flame propagation to the side of uniform distribution.

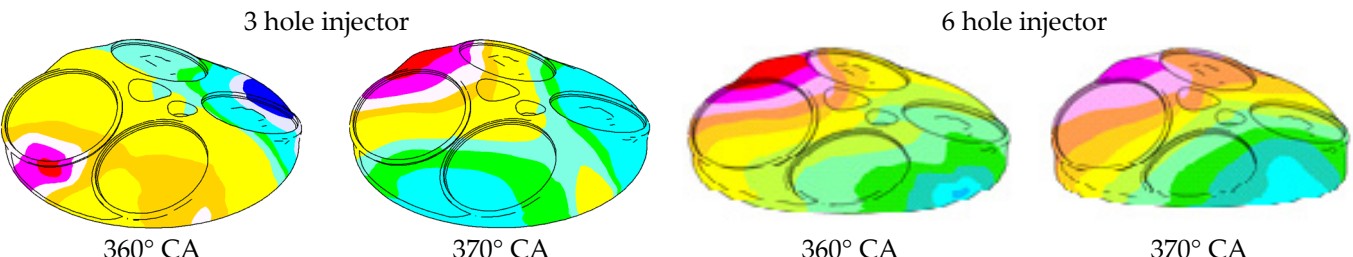

**Figure 10.** Pressure contour at different crank angle position and different injector hole number.

The temperature increment inside the cylinder during the combustion process can be represented in Figure 11. The red region in the illustration represents a highest mixture of temperature around 3430 K for the 3-hole injector and 3180 K for the 6-hole injector. This region shows that the mixture is burning. The phenomenon of the combustion temperature is relatively distributed symmetrically between the side of the intake and exhaust valves, proving that the piston is able to move downward smoothly during the expansion stroke. Therefore, the number of more holes indicates that the combustion moves more slowly due to the uniform distribution of fuel in the combustion chamber.

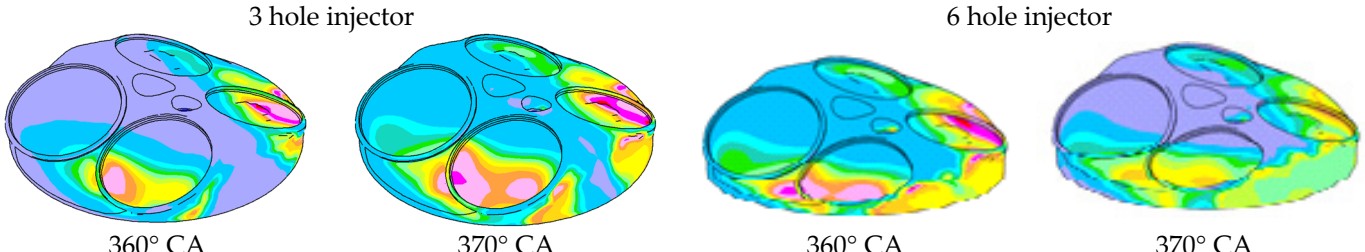

**Figure 11.** Temperature contour at different crank angle position and different injector hole number.

Figure 12 shows the impact of the interaction of two factors—engine speed and hydrogen fuel percentage—on the changes in engine performance parameters. The results confirm that by increasing the engine speed at each level of hydrogen fuel percentage, the power always increases. However, based on the results of Table 7, the interactions are not significant, and therefore, statistically, there is no difference between the trend of power changes versus engine speed along with different levels of hydrogen in the fuel blend. Increasing the engine speed to about 4000 rpm increases the torque and then decreases the torque change trend. The maximum and minimum torque is between 40% and 0% H2 in the fuel blend (HCNG). The changes in FCE, NOx, and CO always trend upward with increasing engine speed, although the highest and lowest amounts of FCE and NOx were obtained with 40% hydrogen and 0% HCNG. In contrast, the highest and lowest CO values

are obtained at 0% and 40% HCNG. Adding hydrogen to the natural gas versus the engine speed will increase the combustion rate of the spark ignition engine, which also improves the thermal efficiency and combustion efficiency [46,47].

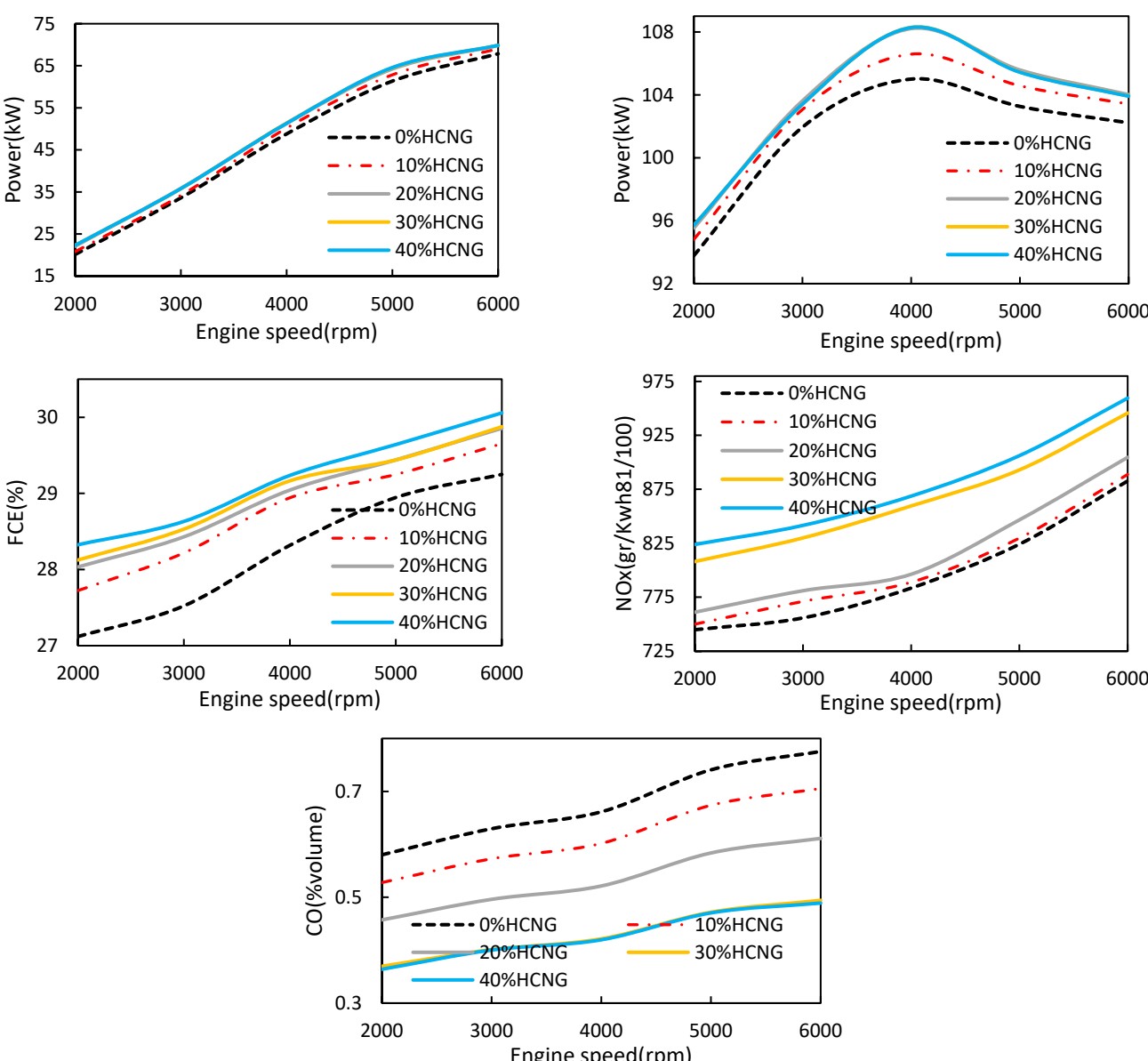

**Figure 12.** Engine performance parameters versus engine speed at different levels of hydrogen fuel content.

The result of this investigation on the interactions of fuel injection pressure (IP) at different engine speeds on changes in engine performance parameters is shown in Figure 13. As the results confirm, by increasing the engine speed at each level of IP, the power always has a significant increasing trend. Increasing the engine speed to approximately 4000 rpm increases the torque and then decreases the torque change trend. Hence, the maximum and minimum power and torque obtained are related to IP = 25 bar and IP = 15 bar. The changes in FCE, NOx, and CO always trend upward with an increase in engine speed at each level of IP. The highest and lowest FCE, and NOx are related to IP = 25 bar and IP = 15 bar. In contrast, the maximum and minimum CO values are obtained at IP = 15 bar and IP = 25 bar, respectively.

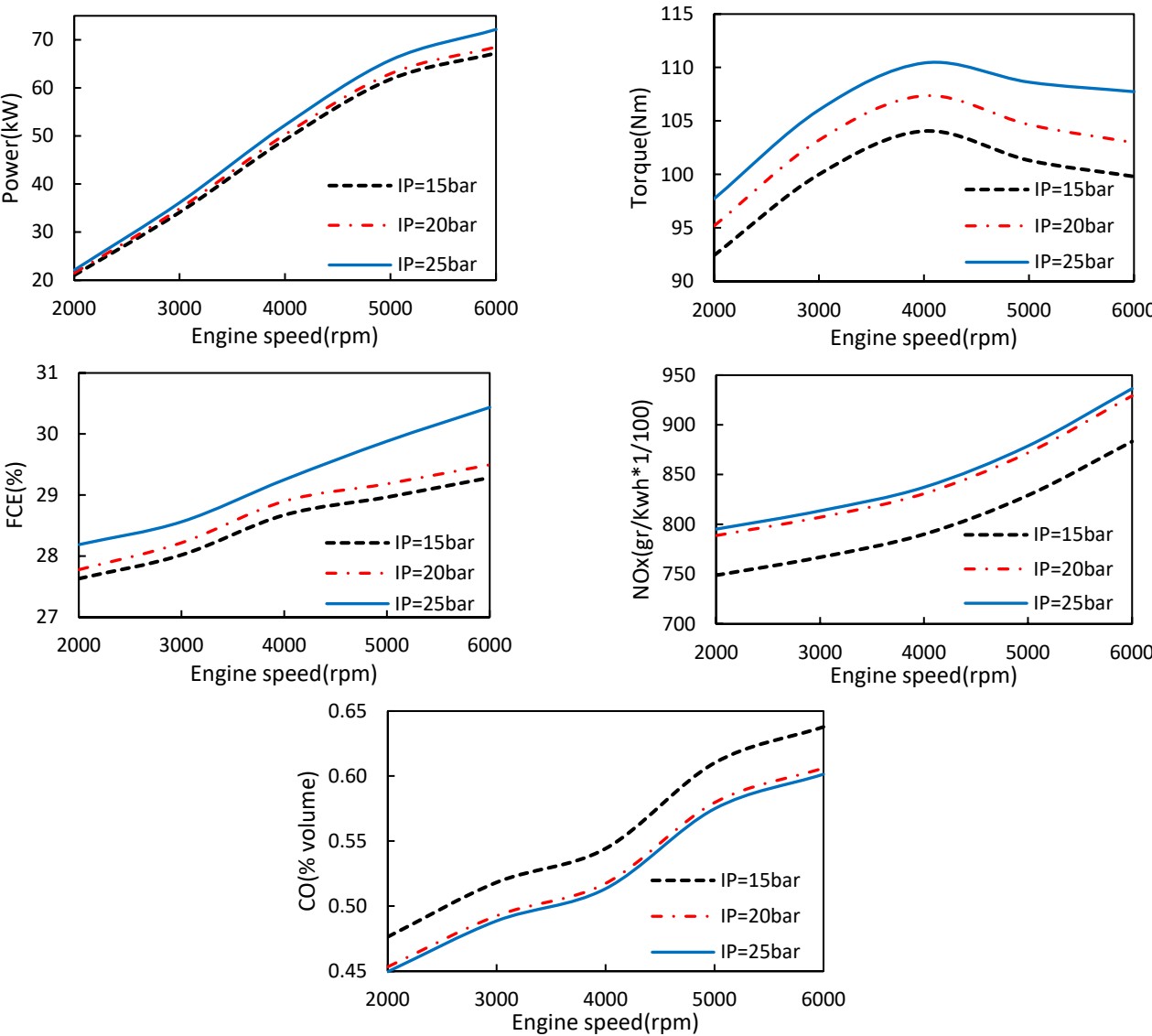

**Figure 13.** The result of changes in engine performance parameters versus engine speed at different levels of fuel injection pressure (IP).

The interaction effects of injector hole number (IHN) at different engine speeds on changes in engine performance parameters are shown in Figure 14. As the results show, by increasing the engine speed at each level of IHN, all engine performance parameters are accompanied by a significant upward trend, although this increasing trend is reversed for torque after 4000 rpm. Hence, based on the results of Table 4, the interactions of injector hole number and engine speed on engine power, torque, and specific fuel consumption are not significant. However, a 3-hole injector has been able to produce more power, torque, and fuel consumption. Increasing the hole number from 3 to 6 has significantly reduced and generated more changes on the NOx and CO. These changes have also become considerable as shown in Table 4.

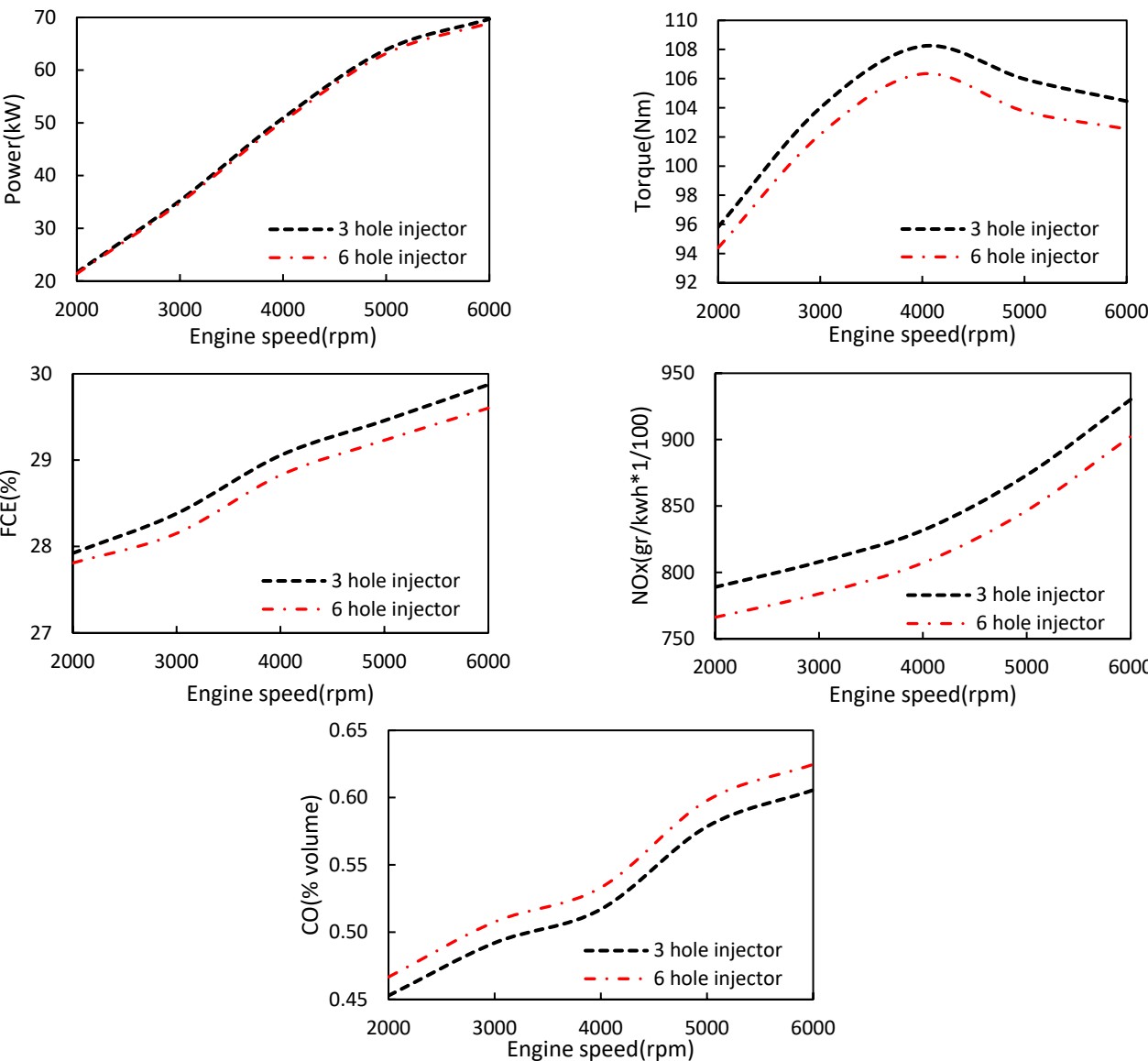

**Figure 14.** The changes of engine performance parameters versus engine speed in respected to injector hole number (IHN).

The results of the interaction effects of hydrogen fuel percentage and injection pressure are shown in Figure 15. As can be seen, power, torque, specific fuel consumption, and NOx have an almost upward trend with an increase in the percentage of hydrogen fuel, although the increment is due to the increase in fuel pressure. The change in CO also shows that as the amount of hydrogen fuel increases, the amount of CO decreases due to complete combustion, but the effect of lost injection pressure was the reverse. Therefore, increasing the injection pressure and also increasing the percentage of hydrogen in the fuel composition has a positive effect on the engine performance. Hence, to obtain a high performance, we need to apply an injection pressure of 25 bar with 30% hydrogen in the fuel blend.

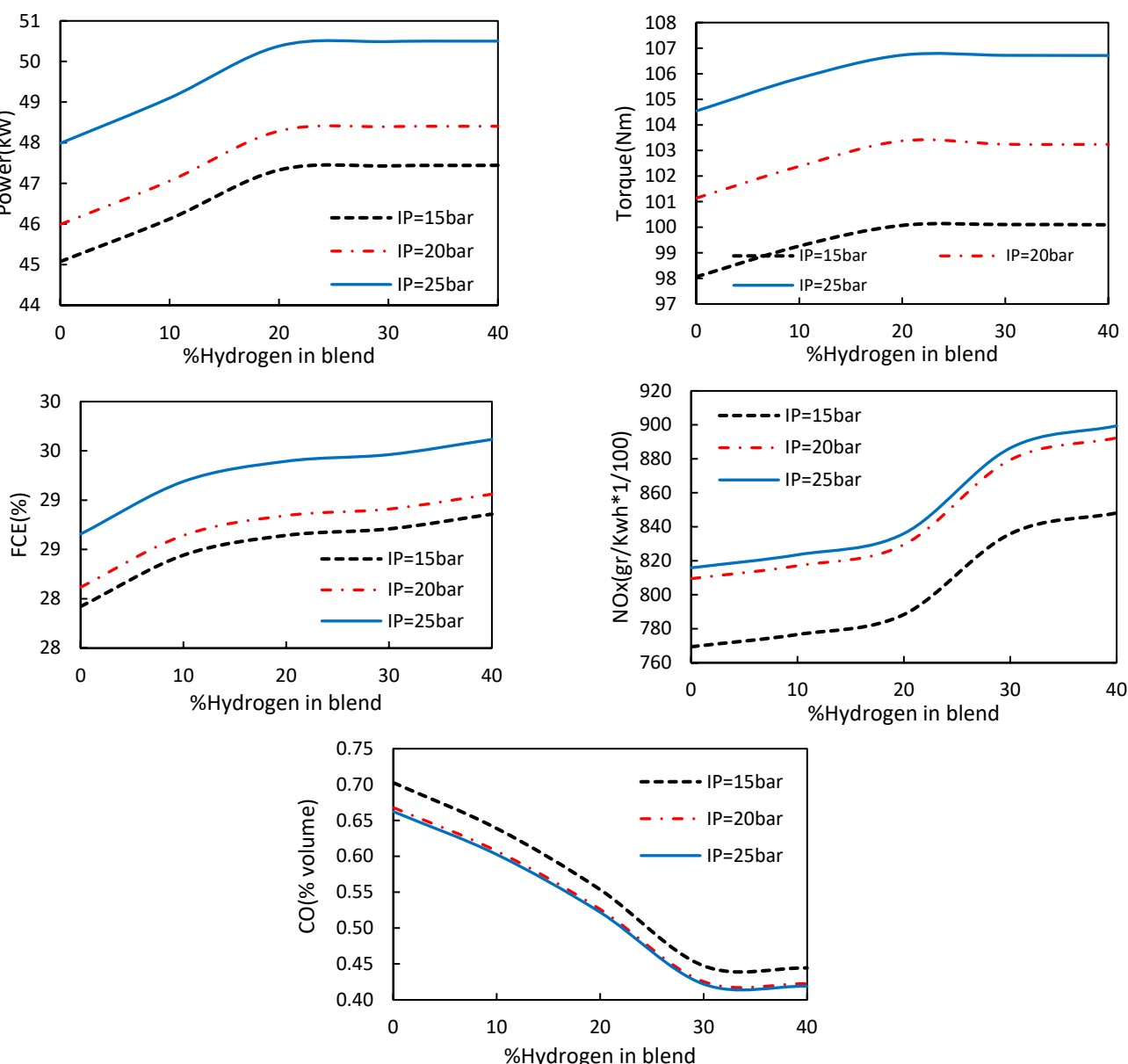

**Figure 15.** The result of changes in engine performance parameters against the percentage of hydrogen penetration at different levels of fuel injection pressure (IP).

The results of the interactions between the percentage of hydrogen in the blend of HCNG and the number of injector holes are shown in Figure 16. As can be seen, power, torque, specific fuel consumption, and NOx have an almost upward trend with an increase of hydrogen. The changes in CO show that as the percentage of hydrogen increases, the amount of CO decreases. The injector holes number, along with the effect of hydrogen fuel, is not significant, and therefore the injector holes number has no significant effect on the changes in power and amount of hydrogen. Hence, more torque is obtained with a greater number of injector holes. This result also corresponds for FCE and $NO_X$.

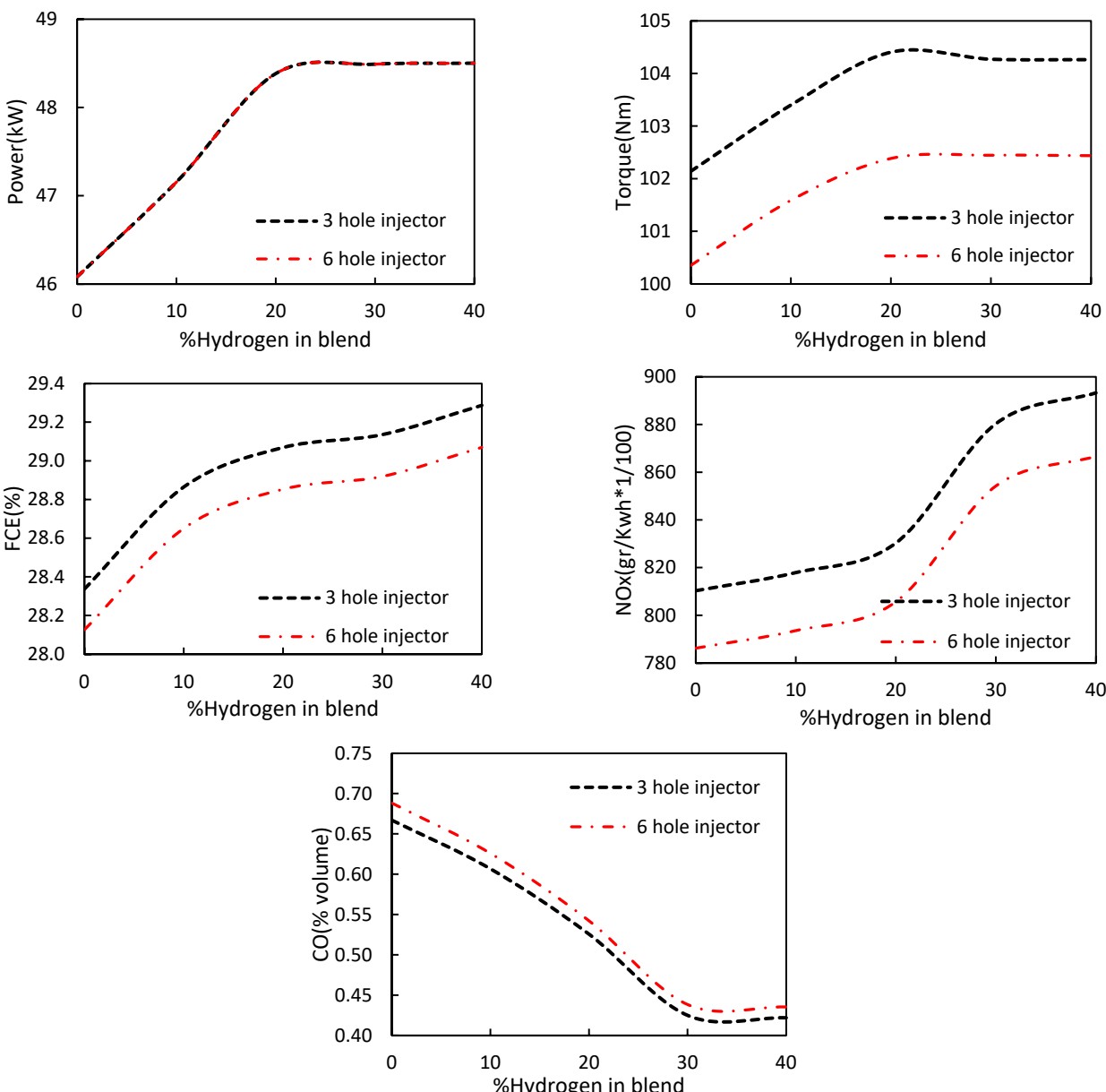

**Figure 16.** The result of changes in engine performance parameters versus the percentage of hydrogen fuel at different levels of the injector hole number (IHN).

Figure 17 shows the result of the interactions of two factors, IP and IHN, for engine performance parameters. According to the following figures, it can be specified that the engine performance with respect to injector hole number and injection pressure, such as power, torque, and fuel efficiency increases with a lower injector hole number and the increment of injection pressure, but according to the two parameters, the optimal values of the engine performance obtained with 21 bar injection pressure and a 6-hole injector. An increase of the injection pressure causes the performance of the engine to practically increase, and this is due to the reduction of fuel viscosity at high injection pressures [48], as well as the atomization of the fuel and finally more complete combustion of the fuel in the combustion chamber. Although CO values decrease with an increase in the injection pressure and a decreasing injector hole number, NOx values show the opposite. The increase in NOx is due to the presence of excess oxygen as well as the high combustion temperature [49].

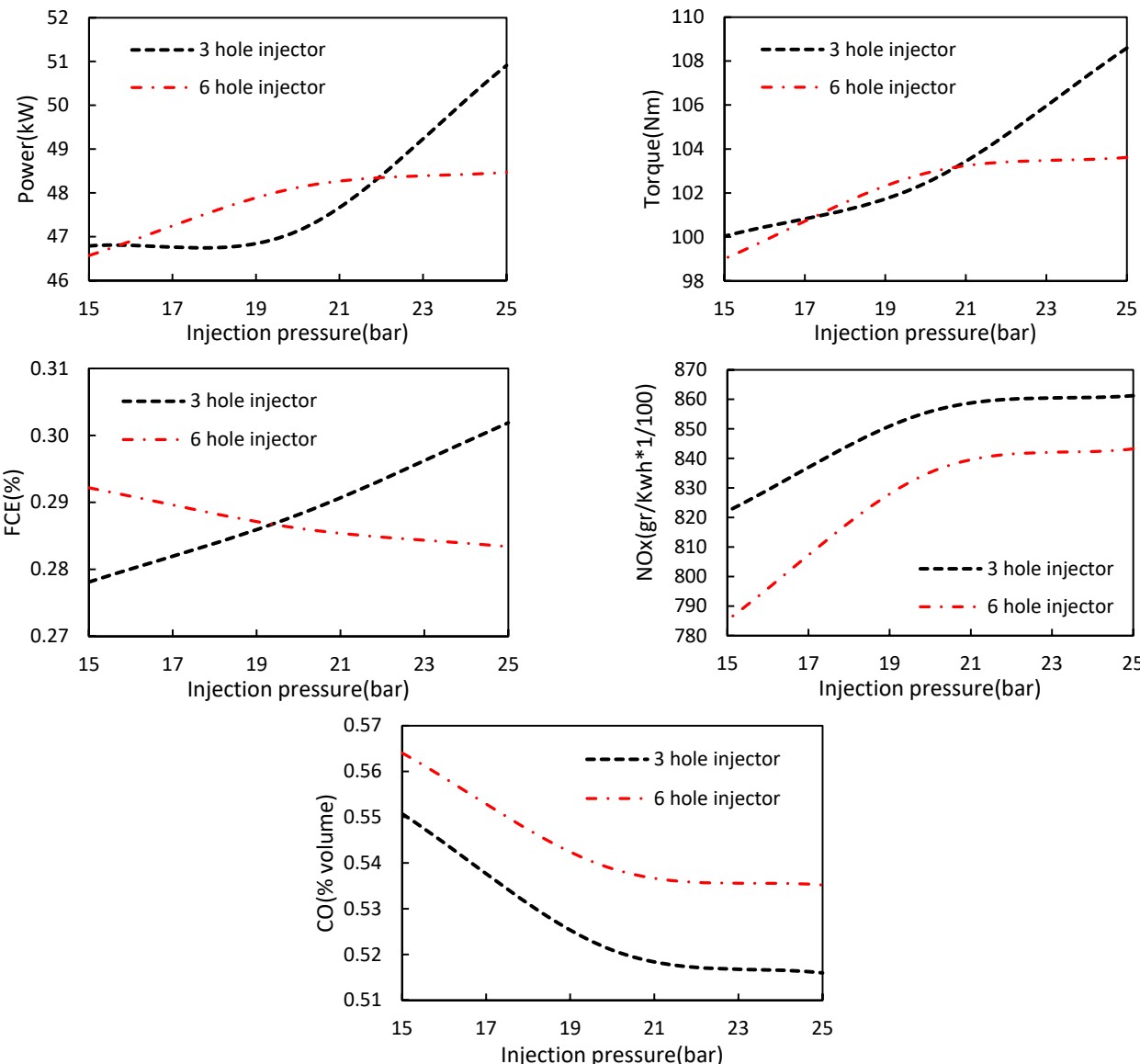

**Figure 17.** Variation of engine performance parameters versus injection pressure (IP) and injector hole number (IHN).

Table 8 shows the optimal values of the engine performance variables along with the exhaust gases based on various engine parameters, including the percentage of excess hydrogen in the fuel blend, injection pressure, and injector hole number at different engine speeds. The results confirm that the optimal value in each engine cycle is different.

**Table 8.** Optimization values of engine performance variables.

| rpm | H% | IP (bar) | IHN | CO (% Volume) | NOx (ppm) | FCE (%) | T (Nm) | P (kW) |
|------|-----|----------|-----|---------------|-----------|---------|--------|--------|
| 2000 | 20  | 25       | 3   | 0.44          | 785.84    | 0.29    | 100.49 | 23.77  |
| 3000 | 20  | 25       | 6   | 0.49          | 790.20    | 0.28    | 104.01 | 35.79  |
| 4000 | 20  | 25       | 6   | 0.52          | 805.39    | 0.28    | 108.77 | 51.71  |
| 5000 | 30  | 15       | 3   | 0.48          | 880.69    | 0.28    | 102.58 | 62.98  |
| 6000 | 20  | 15       | 3   | 0.63          | 891.80    | 0.29    | 100.87 | 67.96  |

## 4. Conclusions

In this study, the effect of various parameters including the amount of hydrogen in the fuel blend, injection pressure, and injector hole number on the performance and emission characteristics of the engine exhaust, and the following findings were obtained.

1.  By increasing the percentage of hydrogen in the fuel composition up to 20%, the changes in power, torque, and fuel consumption efficiency trend significantly upward statistically.
2.  By increasing the percentage of hydrogen in HCNG fuel, NOx increased due to an increase in the temperature of the combustion chamber and also caused a significant decrease in CO values.
3.  By increasing the fuel injection pressure along with the engine speed, the FCE, NOx, and CO values went up, although the maximum power and torque were obtained at an injection pressure of 25 bar and 15 bar.
4.  Reducing the injector hole number at each step increases the performance characteristics of the engine, whereas the amount of torque after 4000 RPM has the opposite trend. At the same time, the amount of NOx and CO pollutants has significantly reduced and generated many changes, respectively, by decreasing the injector hole number from 6 to 3.
5.  Increasing the injector hole number along with the percentage of hydrogen in the HCNG did not have a significant effect on the performance characteristics. Therefore, the amount of CO decreases with an increase in the percentage of hydrogen, and the amount of NOx and FCE with higher hole numbers in the injector had an increasing trend.
6.  The optimal value of engine performance was obtained according to the two parameters of injection pressure at 19 bar and injector hole number of 3 holes.
7.  One general result is that the optimal engine conditions in this research were 30% HCNG fuel, an injection pressure of 25 bar, and an injector with 6 holes. In addition, adding hydrogen to CNG had an axial role in the improvement of engine performance.

**Author Contributions:** Conceptualization, J.Z.; Data curation, J.Z.; Formal analysis, J.Z.; Funding acquisition, J.Z. and J.R.N.A.; Investigation, J.Z. and J.R.N.A.; Methodology, J.Z., J.R.N.A., Y.L.A. and M.R.G.; Project administration, J.Z., J.R.N.A., Y.L.A. and M.R.G.; Resources, J.Z., J.R.N.A., Y.L.A. and M.R.G.; Software, J.Z., J.R.N.A., Y.L.A. and M.R.G.; Supervision, Validation, J.Z., J.R.N.A., Y.L.A., M.R.G. and Á.R.A.L.; Visualization, J.Z., J.R.N.A., Y.L.A., M.R.G. and Á.R.A.L.; Writing—original draft, J.Z., J.R.N.A., Y.L.A. and M.R.G.; Writing—review & editing, J.Z., J.R.N.A., Y.L.A. and M.R.G. All authors have read and agreed to the published version of the manuscript.

**Funding:** This research received no external funding.

**Data Availability Statement:** On each request, we provide the data detail.

**Conflicts of Interest:** The authors declare no conflict of interest.

## Nomenclature

| | |
|---|---|
| DI | Direct Injection |
| CA | Crank Angle |
| LHV | Lower Heating Value |
| RPM | Revolutions Per Minute |
| HCCI | Homogeneous Charge Compression Ignition |
| CRD | Completely randomized design |
| IP | Injection Pressure |
| CR | Injector hole number |
| FCE | Specific Fuel Consumption |
| Ns | Not significant. |
| SS | Total squares |
| IHN | Injector hole number |

Gases and Fuels
NG  Natural Gas
CNG  Compressed Natural Gas
HCNG  Hydrogen enriched Compressed Natural gas
NOx  Nitrogen Dioxide
CO  Carbon Monoxide
$CO_2$  Carbon Dioxide
H  Hydrogen fuel consumption
Greek Letters
P  Density of the fluid flow
$\hat{U}i$  Local velocity
Gi  Acceleration
P  Pressure of the fluid flow
M  Kinematic viscosity
Mi  Kinematic viscosity
Vi  Stress tensor
Vj  Stress tensor
$\Sigma ij$  Stress from interaction of the fluid flow
H  Local enthalpy of the fluid flow
Qg  Heat exchange rate in the gaseous mixture
Tij  Shear stress between the fluid flow lines
1  Fuel ratio
T  Fluid flow temperature
Yi  Introduced species
V  Viscosity of the fluid flow
Ji  Infiltration of the species
Ri  Rate of production of the species
Si  Source of the species

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
