# Peer review of "A Simulation Study of the Effect of HCNG Fuel and Injector Hole Number along with a Variation of Fuel Injection Pressure in a Gasoline Engine Converted from Port Injection to Direct Injection"

_processes, doi:10.3390/pr10112389_

Round 1
Reviewer 1 Report
Author presents an introduction well written and recall certains works in the domain in order to present some old results (37 in total) in the domain. The aim of the work is also presented. Numerical values have been validated used experimental data taken from the literature. Numerical values have been discussed used statistical parameters. All figures and Tables are easy to read. This is why I propose only some minor revisions in order to increase the quality of the paper.
My questions:
*Introduction section is not clear and very long. Can you decrease the number of lines and focus only on main results obtained in tha paper.
*Author used the AVL Fire software in this work, please add some references of old works that used the same tool in this domain.
*Add the references of each color of the fig 1
* Give some explanations why "The selected engine for this investigation is a four-stroke engine with four cylinders, gasoline type port injection then converted to direct injection mode for the fuel conversion to natural gas and hydrogen fuel."
*Also, give all reasons to choice the values given in the tables 1 and 2.
*I don't understand the dimension of Equation 5. Check this equation, I think that it is not well written.
*Explain why you choice values of tables 3 and 4.
Author Response
Processes journal
Manuscript Title: A simulation study of the Effect of HCNG fuel and injector hole number along with a variation of fuel injection pressure in a gasoline engine converted from port injection to direct injection
Manuscript ID: processes-1959079
Dear editor-in-chief, Prof. Dr. Giancarlo Cravotto,
Department of Drug Science and Technology, University of Turin, Via P. Giuria 9, 10125 Turin, Italy
Thank you so much to give me an opportunity to publish my paper in your journal. I received the revised letter with some comments from the editor and reviewers and changed the manuscript according to the reviewer's suggestions and highlighted the changed sentences/words with yellow paint in the manuscript, also I answered the questions from reviewers respectively in this sheet. Thanks a lot for the valuable comments.
Kind regards
- Javad Zareei
Reviewer 1: The author presents an introduction well written and recalls certain works in the domain in order to present some old results (37 in total) in the domain. The aim of the work is also presented. Numerical values have been validated used experimental data taken from the literature. Numerical values have been discussed used statistical parameters. All figures and Tables are easy to read. This is why I propose only some minor revisions in order to increase the quality of the paper.
Comment 1: The introduction section is not clear and very long. Can you decrease the number of lines and focus only on the main results obtained in the paper?
Response: We sincerely appreciate the reviewer’s positive feedback. The modifications have been done as requested.
Comment 2: The author used the AVL Fire software in this work, please add some references to old works that used the same tool in this domain.
Response: Thank you very much. As suggested, we added some references to the introduction.
Comment 3: Add the references of each color of fig 1.
Response: Thanks for your valuable advice, we did it.
Comment 4: Give some explanations why "The selected engine for this investigation is a four-stroke engine with four cylinders, gasoline type port injection then converted to direct injection mode for the fuel conversion to natural gas and hydrogen fuel."
Response: Because this engine was chosen as a research project and we had done some work including converting the port injection engine to direct injection in a practical way, the necessary changes such as the location of the injector and the spark plug were also properly designed by a research team.
Comment 5: Also, give all reasons to choice the values given in the tables 1 and 2.
Response: In Table 1- these data are related to the basic engine and also the optimization of fuel injection time. Table 2 also shows the standard injector data available in the market.
Comment 6: I don't understand the dimension of Equation 5. Check this equation, I think that it is not well written.
Response: I checked the equation and it was correct, but I corrected some items in the next paragraph.
Comment 7: *Explain why you choice values of tables 3 and 4.
Response: Some of them are related to the basic engine, and some data like the injection pressure are related to the chosen injector, and also the data related to the meshing, which is selected based on the methods and type of mesh selection for a better combustion solution.
Reviewer 2 Report
Article describes the study of influence of HCNG fuel content, injector number of holes and direct injection pressure in gasoline converted engine on tested system performance. Amount of hydrogen enrichment by about 30% gave the best results of engine performance. Than 25 bar injection pressure was the most optimal . Also 6 hole injector was the best according to exhaust of harmful gases.
Please add some recent references concerning similar study.
Author Response
Processes journal
Manuscript Title: A simulation study of the Effect of HCNG fuel and injector hole number along with a variation of fuel injection pressure in a gasoline engine converted from port injection to direct injection
Manuscript ID: processes-1959079
Dear editor in chief, Prof. Dr. Giancarlo Cravotto,
Department of Drug Science and Technology, University of Turin, Via P. Giuria 9, 10125 Turin, Italy
Thank you so much to give me an opportunity to publish my paper in your journal. I received the revise letter with some comments from editor and reviewers and changed the manuscript according to the reviewers suggestions and highlighted the changed sentences/words with yellow paint in the manuscript, also I answered the questions from reviewers respectively in this sheet. Thanks a lot for the valuable comments.
Kind regards
- Javad Zareei
Reviewer 2: Article describes the study of influence of HCNG fuel content, injector number of holes and direct injection pressure in gasoline converted engine on tested system performance. Amount of hydrogen enrichment by about 30% gave the best results of engine performance. Than 25 bar injection pressure was the most optimal . Also 6 hole injector was the best according to exhaust of harmful gases.
Comment 1: Please add some recent references concerning similar studies.
Response: We sincerely appreciate the reviewer’s positive feedback. The modifications have been done as requested.